# Low MTUS1 Protein Expression Is Associated with Poor Survival in Patients with Colorectal Adenocarcinoma

**DOI:** 10.3390/diagnostics13061140

**Published:** 2023-03-16

**Authors:** Hosub Park, Seungyun Jee, Hwangkyu Son, Hyebin Cha, Seongsik Bang, Byung Kyu Ahn, Jaekyung Myung, Seungsam Paik, Hyunsung Kim

**Affiliations:** 1Department of Pathology, Hanyang University Hospital, Hanyang University College of Medicine, Seoul 04763, Republic of Korea; 2Department of Surgery, Hanyang University Hospital, Hanyang University College of Medicine, Seoul 04763, Republic of Korea

**Keywords:** microtubule-associated tumor suppressor 1 (MTUS1), prognosis, colorectal adenocarcinoma

## Abstract

Introduction: Microtubule-associated tumor suppressor 1 (MTUS1) is a novel tumor suppressor protein involved in cell proliferation, migration, and tumor growth. MTUS1 is thought to be downregulated in various human cancers and associated with poor prognosis. We evaluated the clinicopathologic significance and prognostic value of MTUS1 in colorectal adenocarcinoma. Methods: Immunohistochemical staining for MTUS1 was performed on tissue microarrays of 393 colorectal adenocarcinoma cases, and MTUS1 staining was classified into high- and low-expression groups. Then, we investigated the correlations between MTUS1 protein expression and various clinicopathological parameters and patient survival. Results: MTUS1 protein was expressed at various grade levels in the cytoplasm of tumor cells, which showed loss or decreased expression of MTUS1. A total of 253 cases (64.4%) were classified into the low MTUS1 protein expression group and 140 cases (35.6%) into the high MTUS1 expression group. A low level of MTUS1 protein significantly correlated with tumor size (*p* = 0.047), histological grade (*p* < 0.001), lymphovascular invasion (*p* < 0.001), perineural invasion (*p* = 0.047), and lymph node metastasis (*p* < 0.001). Survival analyses showed that patients with low MTUS1 protein expression had worse overall survival (*p* = 0.007, log-rank test) and worse recurrence-free survival (*p* = 0.019, log-rank test) than those with high MTUS1 expression. Conclusions: Low MTUS1 protein expression is associated with adverse clinicopathological characteristics and poor survival outcomes in patients with colorectal adenocarcinoma. These results suggest that MTUS1 functions as a tumor suppressor in colorectal adenocarcinoma and could be a potential prognostic biomarker.

## 1. Introduction

According to 2018 global cancer statistics, colorectal cancer is one of the most common cancers, the second-leading cause of cancer-related deaths, and the second- and third-most common malignancy in women and men, respectively [1]. Internationally, colorectal cancer is increasing, mainly in low- and middle-income countries [2]. In addition, early-onset colorectal cancer is increasing in young adults in developed countries such as the United States, Canada, and Australia [3,4,5]. Despite recent outstanding advances in the understanding of carcinogenesis and treatment modalities of colorectal cancer, therapeutic advances are not yet well established. Although there are numerous ongoing studies investigating alternative molecular biomarkers to be used as a novel adjuvant therapy, there is no proper target therapy available to improve outcomes in colorectal cancer patients [6]. 

The microtubule-associated tumor suppressor 1 gene (*MTUS1;* also known as mitochondrial tumor suppressor gene 1) is located on chromosome 8p22. It contains 17 exons that encode a protein with a C-terminal domain that interacts with the angiotensin II (AT2) receptor. The *MTUS1* gene has transcript variants of various isoforms through alternative splicing that encode mitochondrial protein with tumor suppressor activity [7,8]. *MTUS1* is downregulated in several types of cancers, including colorectal carcinoma [9,10], lung adenocarcinoma [11], gallbladder adenocarcinoma [12], renal cell carcinoma [13], bladder carcinoma [14,15], adenoid cystic carcinoma of salivary gland [16], squamous cell carcinoma of the tongue [17], head and neck squamous cell carcinoma [18], uveal melanoma [19], and breast carcinoma [20,21].

Recently, the tumor suppressor function of *MTUS1* has been reported at the mRNA level in colorectal adenocarcinoma [10]; however, the correlation between MTUS1 protein level and its prognostic significance has not been reported. The aim of this study was to discover the prognostic significance of MTUS1 protein expression in patients with colorectal adenocarcinoma. We investigated the clinicopathological significance and prognosis of MTUS1 protein expression in a large cohort of patients with colorectal adenocarcinoma by immunohistochemical staining.

## 2. Materials and Methods

### 2.1. Patients and Tissue Samples

We searched the pathologic repository and found 424 cases of colon and rectum resection specimens that were diagnosed as primary colorectal adenocarcinoma, between January 2005 and December 2010 at Hanyang University Hospital, Seoul, Republic of Korea. Electronic medical records and pathology reports were reviewed for clinicopathological information. Among the TMA slides stained for MTUS1, 31 (7.3%) out of 424 cores were uninterpretable due to dropout or the absence of viable cancer tissue in the core. The cases with these uninterpretable cores were excluded from further analysis. This study was approved by the Institutional Review Board of the Hanyang University Hospital (HYUH 2019-11-008-002), and the requirement for informed consent was waived. All hematoxylin and eosin (H&E)-stained slides of the included cases were reviewed with pathology reports and medical records. The assessed clinicopathological characteristics were patient sex; age; tumor size; histological grade; lymphovascular invasion; perineural invasion; pathological T (pT) stage; pathological N (pN) stage; distant metastasis; MSI status; and patient survival. 

### 2.2. Tissue Microarray (TMA) Construction

The cancer tissues from included cases were extracted and inserted into the TMA block. All H&E-stained slides of included cases were reviewed and the most representative cancer area was selected in each case. In this selected cancer area, a 3-mm-diameter tissue cylinder core was punched out from the formalin-fixed paraffin-embedded donor tissue block. The tumor cores were manually inserted in designated positions of a recipient paraffin block for TMA.

### 2.3. Immunohistochemical Staining

The immunohistochemical staining for MTUS1 was performed on 4-μm-thick prepared sections from the TMA blocks using a fully automated slide preparation Benchmark XT System (Ventana Medical Systems Inc., Tucson, AZ, USA). The primary antibody was a polyclonal rabbit anti-MTUS1 antibody (Aviva, San Diego, CA, USA, 1:100 dilution). The staining was processed according to manufacturer instructions.

### 2.4. Interpretation of Immunohistochemical Staining

MTUS1 protein expression was evaluated under light microscopy by two pathologists (HP and SP) who were blinded to the clinicopathological parameters and patient clinical outcomes. Signal intensity was recorded on a 0 to 3 scale, corresponding to negative, weak, moderate, and strong expression, respectively. The percentage of cells at each intensity was recorded in units of 10% points using the eyeballing method. Representative examples of stain intensity are shown in Figure 1. The H score was calculated by multiplying stain intensity and percentage as follows: H-score = (1 × (% cells 1+) + 2 × (% cells 2+) + 3 × (% cells 3+)). The results of MTUS1 staining were classified into low and high protein expression groups at a cutoff point of 60. The cutoff value was that with the highest Youden index in the receiver operating characteristics (ROC) curve, using disease-free survival. 

### 2.5. Statistical Analysis

The Chi-square and Fisher’s exact tests were used for the statistical analysis of categorical variables, and Student’s *t*-test was used for the statistical analysis of continuous variables. The Kaplan–Meier method with the log-rank test and the Cox proportional hazard ratio model was applied for survival analysis. A *p*-value less than 0.05 was determined to be statistically significant. Statistical analysis was performed using R version 4.1.1.

## 3. Results

### 3.1. Baseline Characteristics of the Patients

The baseline characteristics of included cases are summarized in Table 1. The patient population included 243 male patients (61.8%) and 150 female patients (38.2%). The mean age of patients was 63.8 years. The lesion was located at the proximal colon, which includes the cecum, ascending colon, transverse colon, and splenic flexure, in 84 cases (21.4%); 309 cases (78.6%) were located at the distal colon, which includes the descending colon, sigmoid colon, and rectum. Among the 393 cases, 26 (6.6%) were well differentiated histologically, 209 (53.2%) were moderately differentiated, and 158 (40.2%) were poorly differentiated. Lymphovascular invasion was positive in 220 cases (56.0%). Perineural invasion was positive in 185 cases (47.0%). According to the 8th American Joint Committee on Cancer (AJCC) staging system, pT1 was assessed in 26 cases (6.6%), pT2 in 45 cases (11.5%), pT3 in 245 cases (62.3%), and pT4 (including pT4a and pT4b) in 77 cases (19.6%). pN0 was assessed in 186 cases (47.3%), pN1 in 104 cases (26.5%), and pN2 in 103 cases (26.2%). Distant metastasis was recorded in 30 (7.6%) cases. Information on microsatellite instability (MSI) status was available in 340 cases, 91 of which (23.2%) showed MSI-H status. 

### 3.2. Pattern of MTUS1 Protein Expression

Among the IHC-stained TMA slides, 31 (7.3%) out of 424 cores were uninterpretable due to dropout or the absence of viable cancer tissue in the core. MTUS1 protein was expressed at various grades in the cytoplasm of tumor cells showing a loss or decreased expression of MTUS1 protein. Among 393 cases with interpretable cores, 253 (64.4%) were classified in the lower-expression group for MTUS1 protein, and the remaining 140 (35.6%) were classified in the higher-expression group for MTUS1 protein.

### 3.3. Correlations between MTUS1 Protein Expression and Clinicopathological Factors

The associations between clinicopathological variables and MTUS1 protein expression groups are summarized in Table 2. Compared to the high-expression group, the low-expression group for MTUS1 protein showed significantly larger tumor size (*p* = 0.047), worse histological grade (*p* < 0.001), more lymphovascular (*p* < 0.001) and perineural (*p* = 0.047) invasion, and lymph node metastasis (*p* < 0.001). The low-expression group for MTUS1 protein also showed a tendency toward female predominance, but the results were not statistically significant. Age, T stage, distant metastasis, and MSI status showed no significant association with MTUS1 protein expression.

### 3.4. Correlation between MTUS1 Protein Expression and Patient Survival

The low MTUS1 protein expression group showed significantly worse overall survival (log-rank test, *p* = 0.007) and worse recurrence-free survival (log-rank test, *p* = 0.019). The Kaplan–Meier curves for overall and recurrence-free survival are shown in Figure 2. The 3-year overall survival rate was 83.2% in the MTUS1 high-expression group and 73.4% in the low-expression group. The 5-year overall survival rate was 76.3% in the MTUS1 high-expression group and 64.8% in the low-expression group. The 3-year recurrence-free survival rate was 83.2% in the MTUS1 high-expression group and 74.1% in the low-expression group. The 5-year recurrence-free survival rate was 83.2% in the MTUS1 high-expression group and 70.4% in the low-expression group.

The results of univariate and multivariate analyses on the Cox proportional hazard model for overall survival are shown in Table 3. In univariate analysis, the low MTUS1 protein expression group had a higher risk of death than the high MTUS1 protein expression group (*p* = 0.007). Male sex, older age, larger size, presence of lymphovascular invasion, presence of perineural invasion, higher T stage, presence of lymph nodal metastasis, and presence of distant metastasis also imparted a higher risk. In multivariate analysis, male sex, older age, and presence of perineural invasion continued to show a higher risk. The results of univariate and multivariate analyses on the Cox proportional hazard model for recurrence-free survival are shown in Table 4. In univariate analysis, the low MTUS1 protein expression group showed a higher risk of recurrence than the high MTUS1 protein expression group (*p* = 0.02). The presence of lymphovascular invasion, presence of perineural invasion, higher T stage, presence of lymph nodal metastasis, and presence of distant metastasis also imparted a higher risk. In multivariate analysis, male sex, moderate histologic grade, and higher T stage showed a significantly higher recurrence risk. 

## 4. Discussion

In the present study, we investigated MTUS1 protein expression in 393 cases of colorectal adenocarcinoma and evaluated the associations between MTUS1 protein level and clinicopathological parameters, and overall and recurrence-free survival in patients with colorectal adenocarcinoma. Low MTUS1 protein expression is closely correlated with poor prognostic clinicopathological parameters, including larger tumor size, higher histological grade, lymphovascular invasion, perineural invasion, and lymph node metastasis. Kaplan–Meier survival curves demonstrated a significant effect of a low MTUS1 protein level on both overall and recurrence-free survival of patients with colorectal adenocarcinoma. The low MTUS1 protein expression group showed worse overall survival and worse recurrence-free survival than the high expression group.

*MTUS1* is located on chromosome 8p21.3-22 and is called the microtubule-associated tumor suppressor 1 (*MTUS1*) or mitochondrial tumor suppressor gene 1 (*MTSG1*). The *MTUS1* gene generates six transcription variants through alternative splicing. Among them, five variants encode angiotensin-II type 2 (AT2) receptor-interacting proteins (ATIPs). The protein isoforms are ATIP1; ATIP2; ATIP3a; ATIP3b; and ATIP4 [8]. ATIP1 and ATIP3 mediate cellular apoptotic mechanisms and interfere with growth-promoting signals, affecting the occurrence and progression of cancers [10]. MTUS1 was first identified as a novel tumor suppressor in pancreatic malignancy and in various types of cancers including breast, head and neck, colon, and ovarian cancers, and its downregulation has been confirmed [9]. Recent studies have reported that recombinant overexpression of *MTUS1* inhibited tumor cell proliferation, while reduced *MTUS1* expression was associated with increased cell proliferation of oral squamous cell carcinoma cells, breast cancer cells, and ovarian cancer cells [9]. Reduced *MTUS1* expression is also associated with poor prognosis in lung adenocarcinoma [11], gallbladder adenocarcinoma [12], renal cell carcinoma [13], bladder carcinoma [14,15], adenoid cystic carcinoma of the salivary gland [16], and squamous cell carcinoma of the tongue [17].

Several studies have reported a tumor suppressor effect of MTUS1 protein in various human cancers. Jee et al. investigated the expression of MTUS1 protein in lung adenocarcinoma and found that a low MTUS1 level was significantly associated with higher histological grade, lymphovascular invasion, lymph node metastasis, higher tumor stage, and higher Ki-67 proliferation index, and patients with a low MTUS1 protein level showed a poor prognosis [11]. Sim et al. reported that low MTUS1 protein expression had a poor prognosis in gallbladder carcinoma and renal cell carcinoma [12,13]. Rogler et al. and Xiao et al. investigated MTUS1 protein expression in bladder carcinoma [14,15]. They reported that both low MTUS1 protein expression and *MTUS1* mRNA expression were associated with poor prognosis. Zhao et al. reported that the downregulation of MTUS1 protein was associated with a poor prognosis in salivary adenoid cystic carcinoma [16]. Ding et al. reported that the downregulation of MTUS1 protein was associated with de-differentiation, enhanced proliferation, and poor prognosis in oral squamous cell carcinoma [17]. Mahjabeen et al. reported that *MTUS1* mRNA expression was decreased in head and neck squamous cell carcinoma [18]. Lou et al. reported poor prognosis in a group with low *MTUS1* mRNA expression in uveal melanoma [19]. Rodrigues-Ferreira et al. reported that *ATIP3* mRNA expression was significantly reduced in breast cancer with poor clinical outcome, and that re-expression of *ATIP3* inhibited tumor cell proliferation both in vitro and in vivo [20]. Kara et al. reported that MTUS1 protein expression was decreased in cancer tissues compared to normal tissues in breast cancer [21].

Several mechanisms that cause low expression of MTUS1 protein have been reported. Bozgeyik et al. summarized in their review paper that loss of heterozygosity in the chromosome 8p21.3-22 region where the *MTUS1* gene is located has been reported in various types of cancers [8]. Various nucleotide substitutions have been reported in the coding region of the *MTUS1* gene in hepatocellular carcinoma, and head and neck squamous cell carcinoma [22,23]. In addition, Zuern et al. suggested that hypomethylation of CpNpG islands in the promoter region of the *MTUS1* gene could affect MTUS1 protein deficiency in colon cancers [24].

Recently, the relationship between reduced *MTUS1* mRNA expression and poor prognosis has been reported in colorectal adenocarcinoma. Cheng et al. investigated mRNA expression levels in colorectal adenocarcinoma using data from The Cancer Genome Atlas (TCGA) database and performed qPCR to verify *MTUS1* mRNA expression in 38 additional clinical samples [10]. They reported that the group with low mRNA *MTUS1* expression had poor overall survival. They also reported that *MTUS1* mRNA expression is decreased in cases with tumor grade III or IV and N2 stage compared to grade I or II and N0 stage [10]. However, the clinicopathological and prognostic value of MTUS1 expression at the protein level had not been investigated in colorectal adenocarcinoma.

This study has the limitation of being retrospective and collecting cases from a single center. In addition, TMA cores are a relatively weak representative of the entire lesion, since each case was submitted as a single core to the TMA. However, this study is the first to show that MTUS1 expression at the protein level has a significant prognostic effect in colorectal adenocarcinoma. As a result, this study supports the MTUS1 protein as a tumor suppressor in colorectal adenocarcinoma. This result supports the need for further studies on the development of target therapy for MTUS1 protein. In addition, the results of this study provide evidence that the immunohistochemical staining technique can be applied when MTUS1 protein is used as a biomarker for colorectal adenocarcinoma in the future.

## 5. Conclusions

In conclusion, we investigated MTUS1 protein expression and evaluated the association between MTUS1 expression at the protein level and prognosis in 393 cases of colorectal adenocarcinoma. The results of this study suggest that a low MTUS1 protein level is associated with a poor prognosis.

## Figures and Tables

**Figure 1 diagnostics-13-01140-f001:**
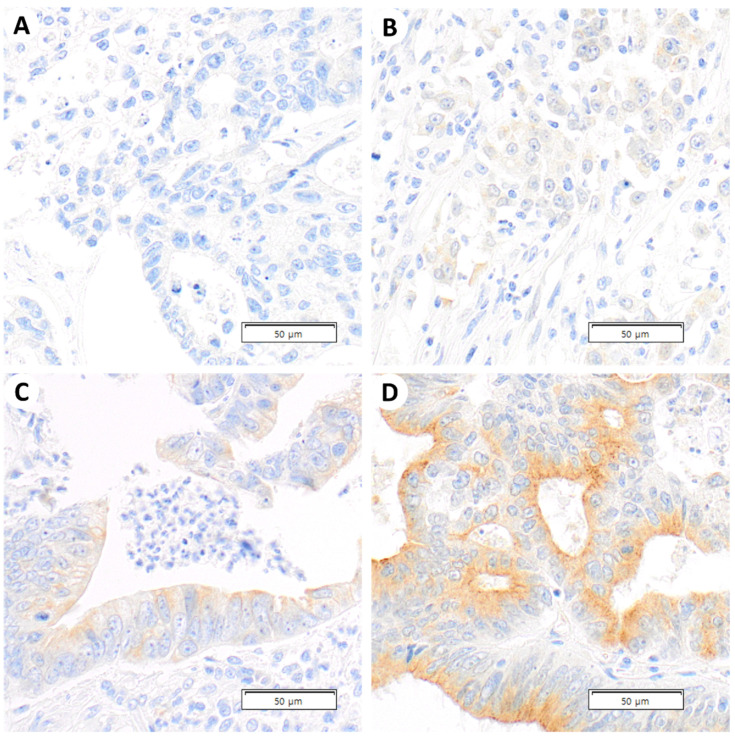
Representative photomicrographs of MTUS1 immunohistochemical stains. (**A**) Negative staining of poorly differentiated tumor cells (×400), (**B**) Weak positivity of poorly differentiated tumor cells (×400), (**C**) Moderate positivity of moderately differentiated tumor cells (×400), and (**D**) Strong positivity of well differentiated tumor cells (×400).

**Figure 2 diagnostics-13-01140-f002:**
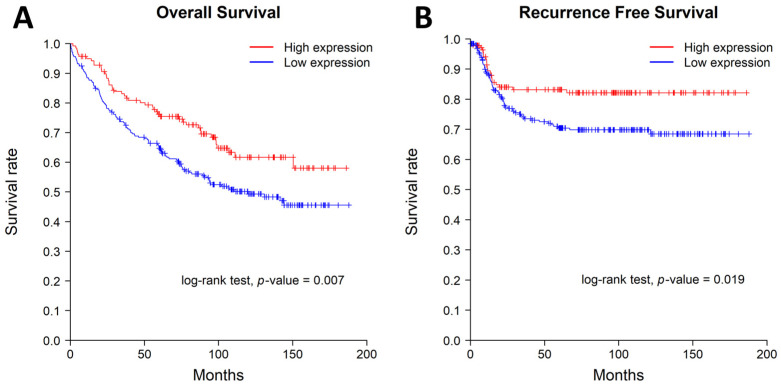
Kaplan–Meier curves for overall survival (**A**) and recurrence-free survival (**B**). The patients with low cytoplasmic MTUS1 protein expression showed worse overall survival (*p* = 0.007) and recurrence-free survival (*p* = 0.019).

**Table 1 diagnostics-13-01140-t001:** Baseline characteristics of enrolled patients (*n* = 393).

Clinicopathological Characteristics	Value (%)
Sex	
Male	243 (61.8%)
Female	150 (38.2%)
Age (years, mean ± SD)	63.8 ± 11.1
Size (cm, mean ± SD)	5.0 ± 2.1
Location	
Proximal	84 (21.4%)
Distal	309 (78.6%)
Histological grade	
WD	26 (6.6%)
MD	209 (53.2%)
PD	158 (40.2%)
Lymphovascular invasion	
Negative	173 (44.0%)
Positive	220 (56.0%)
Perineural invasion	
Negative	208 (53.0%)
Positive	185 (47.0%)
pT stage	
T1	26 (6.6%)
T2	45 (11.5%)
T3	245 (62.3%)
T4	77 (19.6%)
pN stage	
N0	186 (47.3%)
N1	104 (26.5%)
N2	103 (26.2%)
Distant metastasis	
Negative	363 (92.4%)
Positive	30 (7.6%)
MSI status †	
Stable	315 (92.7%)
High	25 (7.3%)

SD, Standard deviation; WD, Well differentiated; MD, Moderately differentiated; PD, Poorly differentiated; MSI, Microsatellite instability; † MSI status was available in 340 cases.

**Table 2 diagnostics-13-01140-t002:** Correlations between MTUS1 expression and clinicopathological parameters (*n* = 393).

Parameter	MTUS1 Expression		*p*-Value
	High (*n* = 140) No. (%)	Low (*n* = 253) No. (%)	
Sex			0.053
Male	96 (68.6%)	147 (58.1%)	
Female	44 (31.4%)	106 (41.9%)	
Age (years, mean ± SD)	63.4 ± 10.5	64.1 ± 11.4	0.554 †
Size (cm, mean ± SD)	4.8 ± 1.9	5.2 ± 2.2	0.047 †
Location			1
Proximal	30 (21.4%)	54 (21.3%)	
Distal	110 (78.6%)	199 (78.7%)	
Histological grade			<0.001
WD	13 (9.3%)	13 (5.1%)	
MD	93 (66.4%)	116 (45.8%)	
PD	34 (24.3%)	124 (49%)	
Lymphovascular invasion			<0.001
Negative	86 (61.4%)	87 (34.4%)	
Positive	54 (38.6%)	166 (65.6%)	
Perineural invasion			0.047
Negative	84 (60%)	124 (49%)	
Positive	56 (40%)	129 (51%)	
T stage			0.356
T1	13 (9.3%)	13 (5.1%)	
T2	18 (12.9%)	27 (10.7%)	
T3	84 (60%)	161 (63.6%)	
T4	25 (17.9%)	52 (20.6%)	
N stage			<0.001
N0	90 (64.3%)	96 (37.9%)	
N1	33 (23.6%)	71 (28.1%)	
N2	17 (12.1%)	86 (34%)	
Distant metastasis			0.941
Negative	130 (92.9%)	233 (92.1%)	
Positive	10 (7.1%)	20 (7.9%)	
MSI status ‡			0.553
Stable	101 (91%)	214 (93.4%)	
High	10 (9%)	15 (6.6%)	

SD, Standard deviation; WD, Well differentiated; MD, Moderately differentiated; PD, Poorly differentiated; MSI, Microsatellite instability; † student *t*-test. ‡ MSI status was available in 340 cases.

**Table 3 diagnostics-13-01140-t003:** Univariate and multivariate Cox regression analysis for overall survival in patients with colorectal adenocarcinoma (*n* = 393).

	Univariate Analysis			Multivariate Analysis		
	HR	95% CI	*p*-Value	HR	95% CI	*p*-Value
Male (vs. Female)	1.455	1.049~2.02	0.025	1.534	1.083~2.172	0.016
Age (per 1 year)	1.058	1.041~1.075	<0.001	1.063	1.045~1.081	<0.001
Size (per 1 cm)	1.103	1.029~1.182	0.006	1.013	0.93~1.103	0.774
Histologic grade WD	1			1		
Histological grade MD (vs. WD)	0.99	0.512~1.913	0.976	0.842	0.406~1.746	0.644
Histological grade PD (vs. WD)	1.545	0.801~2.981	0.195	1.027	0.494~2.137	0.943
Lymphovascular invasion	1.818	1.317~2.509	<0.001	1.517	0.537~4.285	0.432
Perineural invasion	2.548	1.42~4.575	0.002	1.478	1.004~2.174	0.048
T stage T3 & T4 (vs. T1 & T2)	2.073	1.286~3.342	0.003	1.252	0.706~2.22	0.441
Lymph node metastasis	1.863	1.357~2.558	<0.001	0.78	0.28~2.178	0.636
Distant metastasis	2.026	1.268~3.235	0.003	1.417	0.801~2.508	0.231
MSI high (vs. Stable)	0.932	0.491~1.769	0.828	1.447	0.72~2.91	0.3
MTUS1 low expression	1.59	1.132~2.232	0.007	1.4	0.95~2.062	0.089

HR, hazard ratio; CI, confidence interval; vs., versus; WD, Well differentiated; MD, Moderately differentiated; PD, Poorly differentiated; MSI, Microsatellite instability.

**Table 4 diagnostics-13-01140-t004:** Univariate and multivariate Cox regression analysis for recurrence-free survival in patients with colorectal adenocarcinoma (*n* = 393).

	Univariate Analysis			Multivariate Analysis		
	HR	95% CI	*p*-Value	HR	95% CI	*p*-Value
Male (vs. Female)	1.544	0.985~2.422	0.058	2.051	1.254~3.355	0.004
Age (per 1 year)	1.003	0.984~1.023	0.734	1.008	0.986~1.029	0.494
Size (per 1 cm)	1.07	0.974~1.176	0.159	0.945	0.84~1.063	0.346
Histologic grade WD	1			1		
Histological grade MD (vs. WD)	0.936	0.367~2.386	0.89	0.354	0.127~0.988	0.047
Histological grade PD (vs. WD)	1.907	0.76~4.784	0.169	0.664	0.248~1.777	0.415
Lymphovascular invasion	3.091	1.897~5.036	<0.001	2.741	0.691~10.875	0.151
Perineural invasion	2.192	1.433~3.353	<0.001	1.238	0.733~2.092	0.424
T stage T3 & T4 (vs. T1 & T2)	3.166	1.464~6.848	0.003	3.931	1.318~11.722	0.014
Lymph node metastasis	3.099	1.93~4.974	<0.001	0.789	0.211~2.949	0.724
Distant metastasis	3.581	2.084~6.152	<0.001	1.421	0.723~2.794	0.308
MSI high (vs. Stable)	1.028	0.447~2.363	0.947	1.422	0.564~3.587	0.456
MTUS1 low expression	1.75	1.091~2.809	0.02	1.63	0.932~2.852	0.087

HR, hazard ratio; CI, confidence interval; vs., versus; WD, Well differentiated; MD, Moderately differentiated; PD, Poorly differentiated; MSI, Microsatellite instability.

## Data Availability

This study did not use any public databases. The database used in this study is anonymized medical records. However, the authors are unable to disclose the database, because the public release of the database is not approved by the Institutional Review Board.

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
