# Peer review of "Low MTUS1 Protein Expression Is Associated with Poor Survival in Patients with Colorectal Adenocarcinoma"

_diagnostics, 2023, doi:10.3390/diagnostics13061140_

Round 1

Reviewer 1 Report

The article "Low MTUS1 Protein Expression is Associated with Poor Survival in Patients with Colorectal Adenocarcinoma" is fascinating. It explores the protein expression of MTUS1 protein in 393 colorectal adenocarcinoma tissues and evaluates its value as a tumor suppressor gene about different clinical characteristics associated with tumor progression. It is a bit of explored protein. The closest record to this analysis was the evaluation in databases such as TCGA, where gene expression changes and their prognostic value are explored.

Although it has been analyzed in different cáncer types, the information seems highly relevant. The data analyzed starting from the analysis of a microarray of tissues analyzed by immunohistochemistry; I suggest a better quality and organization of the representative images of the microarray that support the findings raised in the proposal, grouped according to the significant clinical variables since the idea seems to be focused on the criteria that determined the categorization of the results, considering size bars and the representative clinical characteristics.

The discussion does not address the functional effects of MTUS1 underexpression related to its role as a tumor suppressor gene, including data on epigenetic regulation, polymorphisms, etc.

Finally, the presentation of the references should be homogenized according to the journal's format.

Reviewer 2 Report

In this manuscript, the authors analyzed the protein expression of MTUS1 in a cohort of colorectal adenocarcinoma by IHC and investigated their correlations with clinicopathological parameters and patient survival. In conclusion, they find that low MTUS1 protein expression is associated with adverse clinicopathological characteristics and poor survival outcomes in patients with colorectal adenocarcinoma. The current manuscript is in a good shape, scholarly presentation is in a right form. However, the MTUS1 expression    and its correlation with clinicopathological parameters and patient survival has been investigated at mRNA, which make this study less interesting. Another problem is that the correlations of MTUS1 protein expression and clinicopathological parameters showed some difference with the correlations of clinicopathological parameters and patient survival based on the results, but it is not mentioned or discussed in the manuscript. Maybe the authors can also investigate the the correlations of MTUS1 protein expression and patient survival in subgroups with different clinicopathological parameters.    
